Impact of urbanization on functional diversity in macromycete communities along an urban ecosystem in Southwest Mexico

Gómez-Hernández Marko 1
Avendaño-Villegas Emily 2
Toledo-Garibaldi María 3
Gándara Etelvina 4 etelvina.gandara@correo.buap.mx
1 CONACYT-CIIDIR Unidad Oaxaca, Instituto Politécnico Nacional , Santa Cruz Xoxocotlán, Oaxaca , Mexico
2 CIIDIR Unidad Oaxaca, Instituto Politécnico Nacional , Santa Cruz Xoxocotlán, Oaxaca , Mexico
3 Institute of Forestry and Conservation, University of Toronto , Toronto , Canada
4 Facultad de Ciencias Biológicas, Benemérita Universidad Autónoma de Puebla , Puebla, Puebla , Mexico
Baird Donald
Electronic publication date: 2021 Sep 21
Publication date: 2021
Volume: 9
Electronic Location ID: e12191
Received 2021 Mar 29; Accepted 2021 Aug 30
Copyright: © 2021 Gómez-Hernández et al.
Copyright year: 2021
Copyright holder: Gómez-Hernández et al.
License: This is an open access article distributed under the terms of the Creative Commons Attribution License, which permits unrestricted use, distribution, reproduction and adaptation in any medium and for any purpose provided that it is properly attributed. For attribution, the original author(s), title, publication source (PeerJ) and either DOI or URL of the article must be cited.
License URL: https://creativecommons.org/licenses/by/4.0/

Keywords: Functional diversity, Macromycete communities, Urban ecosystem, Species richness, Oak forest, South Mexico, Urbanization, Species composition, Functional guilds

Funding: CONACyT 423842 Emily Avendaño-Villegas was supported by the CONACyT scholarship No. 423842. There was no additional external funding received for this study. The funders had no role in study design, data collection and analysis, decision to publish, or preparation of the manuscript.

==============================
Macromycetes are a group of fungi characterized by the production of fruit bodies and are highly relevant in most terrestrial ecosystems as pathogens, mutualists, and organic matter decomposers. Habitat transformation can drastically alter macromycete communities and diminish the contribution of these organisms to ecosystem functioning; however, knowledge on the effect of urbanization on macrofungal communities is scarce. Diversity metrics based on functional traits of macromycete species have shown to be valuable tools to predict how species contribute to ecosystem functionality since traits determine the performance of species in ecosystems. The aim of this study was to assess patterns of species richness, functional diversity, and composition of macrofungi in an urban ecosystem in Southwest Mexico, and to identify microclimatic, environmental, and urban factors related to these patterns in order to infer the effect of urbanization on macromycete communities. We selected four oak forests along an urbanization gradient and established a permanent sampling area of 0.1 ha at each site. Macromycete sampling was carried out every week from June to October 2017. The indices used to measure functional diversity were functional richness (FRic), functional divergence (FDig), and functional evenness (FEve). The metric used to assess variation of macrofungal ecological function along the study area was the functional value. We recorded a total of 134 macromycete species and 223 individuals. Our results indicated a decline of species richness with increased urbanization level related mainly to microclimatic variables, and a high turnover of species composition among study sites, which appears to be related to microclimatic and urbanization variables. FRic decreased with urbanization level, indicating that some of the available resources in the niche space within the most urbanized sites are not being utilized. FDig increased with urbanization, which suggests a high degree of niche differentiation among macromycete species within communities in urbanized areas. FEve did not show notable differences along the urbanization gradient, indicating few variations in the distribution of abundances within the occupied sections of the niche space. Similarly, the functional value was markedly higher in the less urbanized site, suggesting greater performance of functional guilds in that area. Our findings suggest that urbanization has led to a loss of macromycete species and a decrease in functional diversity, causing some sections of the niche space to be hardly occupied and available resources to be under-utilized, which could, to a certain extent, affect ecosystem functioning and stability.

Introduction

Urbanization alters biogeochemical cycles worldwide, contributes to the loss of biodiversity, and has been identified as one of the main causes of species extinctions (McKinney, 2006; Barrico et al., 2012). A broad definition by Niemelä (1999) states that urbanization is the process leading to the increase of densely populated areas characterized by industrial, business, and residential districts. For the past seven decades there has been an exponential growth of urban areas due to population growth and migration (UN, 2018). Nowadays, ca. 4.2 billion people live in urban areas worldwide, and this trend is expected to continue, estimating that by 2,045 there will be 6 billion people living in cities, representing almost 70% of the world’s population (UN, 2014, 2018; McPhearson et al., 2016). Currently, 82% of the population in Latin America is concentrated in urban areas and megacities (i.e., urban areas with more that 10 million people), and Mexico follows the same pattern with 83% of its population of ca. 130 million people living in urban areas (MacGregor-Fors & Ortega-Álvarez, 2013; Worldometers.info, 2020).

Urban areas have been recognized as urban ecosystems composed by anthropogenically built environments and natural or semi-natural areas containing high proportions of non-native plant and animal species (McDonnell & Pickett, 1990; Cousins et al., 2003). Studies in urban areas worldwide have shown that urbanization causes declines in the number of native species and there is a decrease of human influence on diversity from city centers out towards wild areas. Thus, communities of groups like birds, plants, insects, and fungi within cities differ from surrounding communities in non-urban areas (Niemelä, 1999; Grimm et al., 2008; MacGregor-Fors et al., 2015; Avis et al., 2016). Specifically, fungal communities are highly susceptible to habitat loss and can be extremely affected by urbanization, producing a decrease in species richness and changes in species composition among habitat conditions, determined mainly by microclimate, atmospheric pollutants, and factors related to substratum availability and vegetation structure (Newbound, Mccarthy & Lebel, 2010; MacGregor-Fors et al., 2015; Avis et al., 2016; Gómez-Hernández, Ramírez-Antonio & Gándara, 2019).

Fungi comprise one of the most diverse groups in the world (ca. 5.1 million of species) and include both micromycetes (microscopic fungi) and macromycetes (macroscopic fungi) (Blackwell, 2011). Macromycetes are fungi that produce fruit bodies visible to the naked eye and are of vital importance in most terrestrial non-urban and urban ecosystems worldwide (Dighton, 2003). Pathogenic macromycetes can act as natural regulators of plant and animal populations, thus affecting the structure of communities and species productivity (Hansen & Stone, 2005; Deacon, 2006). Saprotrophic and wood-decaying macrofungal species are the main organisms involved in the degradation of organic matter and play a fundamental role in soil formation and nutrient cycling (Lodge & Cantrell, 1995; Mueller & Bills, 2004; Deacon, 2006; Lonsdale, Pautasso & Holdenrieder, 2008). Ectomycorrhizal fungi establish symbiotic associations with plants, facilitating plant uptake of nutrients such as phosphorus and nitrogen in exchange for photosynthetically fixed carbon (Hall, Yun & Amicucci, 2003; Egli, 2011). Nevertheless, macromycetes are highly vulnerable to environmental and microclimatic changes, and habitat modification can impact macrofungal diversity, distribution, and fruit body production (O’Dell, Ammirati & Schreiner, 1999; Brown, Bhagwat & Watkinson, 2006; Durall et al., 2006).

Evaluating variations in species richness and composition along urban landscapes is of importance to identify anthropogenic factors affecting groups of organisms, like macromycetes, that can be of great importance for ecosystems, but susceptible to environmental changes (Deacon, 2006). It is widely known that species have different roles in ecosystem processes and respond differently to environmental pressures due to their particular set of functional traits. Traditional diversity metrics based solely on the number of species can be uninformative because all species are considered as equivalent units (Diaz & Cabido, 2001; Hooper et al., 2002; Flynn et al., 2009). Consequently, there is wide interest in incorporating metrics based on functional traits to research on species diversity and ecosystem functionality (Crowther et al., 2014; Koide, Fernandez & Malcolm, 2014; Aguilar-Trigueros et al., 2015; Pringle, Vellinga & Peay, 2015; Caiafa et al., 2017). Functional traits are quantifiable, morphological, biochemical, physiological, phenological or reproductive characteristics which affect the performance or fitness of organisms under different environmental conditions (response traits) or determine how species contribute to ecosystem functionality (effect traits) (McGill et al., 2006; Cadotte et al., 2009; Tsianou & Kallimanis, 2016; Caiafa et al., 2017). Although there is a recognized need to assess macromycete diversity patterns using a trait-based approach and to understand how urbanization affects macrofungal communities, studies on macromycete functional diversity are scarce. This is particularly true for urban ecosystems where the fungal diversity may be compromised, and therefore, also their role as mutualists, pathogens, and organic matter decomposers within native vegetation remnants and green spaces in urbanized areas (Newbound, Mccarthy & Lebel, 2010).

In Mexico, the state of Oaxaca is the most biologically and culturally diverse region. Currently, 77% of Oaxaca’s ca. 3.9 million people live in cities, and estimates indicate a population increase of 10.9% by 2,030 (Flores-Villela & Gerez, 1994; CONAPO, 2014; INEGI, 2015b). In this context, the aims of our study were to evaluate patterns of macromycete species richness, diversity, functional diversity, and species composition along an urban ecosystem in a heterogeneous landscape in central Oaxaca, Mexico, and to identify the microclimatic, environmental and urban factors related to these patterns. We related these aims to the following hypotheses: (1) patterns of species richness and diversity are positively correlated, and both metrics decrease with increasing urbanization, (2) humidity and temperature are the main drivers of species richness variation along the urbanization gradient, and also are related with the turnover of species composition, (3) the composition of macromycete species highly differs between the most urbanized and the least urbanized sites, (4) urbanization variables negatively affect functional diversity, and (5) different types of functional traits display different patterns along the urbanization gradient.

Materials & Methods

Study area and sites

The study was conducted in the region of Valles Centrales in Oaxaca, Mexico. This region is found in central Oaxaca, between 18° 40′–15° 39′ N and 93° 52′–98° 33′ W, with an elevation ranging from 1,010 to 2,600 m. The region encompasses 121 municipalities and covers an area of ca. 12,000 km2. The mean annual temperature in Valles Centrales is 22 °C, and mean annual precipitation is 1,550 mm (INEGI, 2017).

Three oak forest areas were selected in localities surrounding Oaxaca city: Benito Juarez National Park (Site 1) at 2,254 masl, Las Antenas (Site 2) at 1,711 masl, Santo Domingo Tomaltepec (Site 3) at 1,678 masl; and the Cerro el Fortín Ecological Reserve (Site 4), an area within Oaxaca city at 1,769 masl. At each site, two 100 × 20 m areas were delimited at least 30 m away from the forest edge. Within each of these areas, five permanent 10 × 10 m plots were haphazardly established to cover a sampling area of 0.1 ha.

Authorization to carry out field work in Cerro del Fortin was given by Directorate of Natural Resources and Sustainable Development of Oaxaca (approval number DG/DRNDS/DOETANP/2985/2015). Authorizations in Benito Juarez National Park, Las Antenas, and Santo Domingo Tomaltepec were given verbally by local authorities.

Explanatory variables

A total of 15 microclimatic, environmental, vegetation and urbanization variables were used as explanatory variables in this study. Each sampling day, air humidity (%), air temperature (°C), soil humidity (%), and soil temperature (°C) at the center of each plot were recorded at the four study sites. Slope (°), and soil and stoniness cover (%) were measured once at every plot. Litter depth (cm) was recorded in each plot at the beginning, middle, and end of the sampling season. Woody plants with a diameter >5 cm at 1.3 m above the ground were identified and counted, and their diameter and height were measured. These latter data were used to calculate three variables representing vegetation structure which served as explanatory variables: density (individuals ha−1), basal area (m2 ha−1), and mean tree height (m).

To describe the urbanization gradient along our study area, we selected eight variables as indicators of land-use intensity and disturbance. Urbanization gradients are typically represented by a transect running from an urban core out to rural hinterlands, by measures of population density (Conway & Hackworth, 2007), land cover (McDonnell & Hahs, 2008), or land use factors (Ortega-Álvarez, Rodríguez-Correa & MacGregor-Fors, 2011). In this study we included measures of land cover and population density. Urbanization variables were screened for collinearity using a Pearson correlation matrix to identify any high correlations among them. After removing the correlated variables, we retained five: extent of built area (m2), extent of crop area (m2), population density (individuals ha−1), extent of areas with introduced vegetation (m2), and street and road length (m). To determine the urbanization gradient in the study area, the explanatory variables were measured within a circular buffer of 3 km radius from the center of each sampling site. The 3 km radius was selected to allow a considerable distance among buffer boundaries. All urbanization variables were derived from land-use data layers and census data obtained from the National Institute of Statistics and Geography (INEGI, 2013; INEGI, 2015a), and the total values of the variables were calculated per buffer using zonal statistics in ArcGIS 10.4 (Esri, Redlands, CA, USA).

Macromycete sampling, species richness, diversity and species composition

Sampling was carried out weekly in each plot at every study site throughout the rainy season (June to October) of 2017. The criterion used for macromycete sampling was to collect all the fruit bodies visible to the naked eye. Fruit bodies of the same species within a 50 cm radius, caespitose growth, fairy rings, and fruit bodies of the same species growing on the same log or branch were recorded as a single collection unit, and abundance was estimated as the number of collection units (adapted from Schmit, Murphy & Mueller, 1999). Hereafter, for practical purposes, collection units will be referred to as individuals. Specimens were separated at the species level based on their micromorphological and macromorphological characteristics, and unidentified taxa were classified as numbered morpho-species using a higher taxonomic level approach. Specimens were identified using taxonomic keys and guides (Guzmán, 1977; Largent & Thiers, 1977; Largent, Johnson & Watling, 1977; Breitenbach & Kränzlin, 1984; Breitenbach & Kränzlin, 1986; Largent, 1986; Pegler, 1986; Largent & Baroni, 1988; Breitenbach et al., 1991).

Species richness (Sobs) was estimated as the total number of species observed in each study site. Macrofungal diversity was calculated with the true diversity index of first order (1D) using the multiplicative diversity decompositions of the effective number of species (Jost, 2006; Jost, 2007) by means of the entropart package in R v. 3.4.2 (R Development Core Team, 2017). The species richness estimator Jacknife 2 was used to determine how complete were the inventories of species recorded in every study site, and the turnover of species composition among study sites was assessed with the Chao-Jaccard similarity index (Chao et al., 2005). Both calculations were performed in EstimateS 9.1.0 (Colwell, 2013).

Functional diversity and functional value indexes

To calculate functional diversity of macromycetes, we selected five morphology-related functional traits and arranged them into three trait groups: (1) morphological traits (fruit body texture, hymenium type, pileus diameter), (2) reproductive traits (spore size and shape), and (3) all the traits together (Caiafa et al., 2017). Texture was classified as delicate, jelly-like, fleshy, gristly, coriaceous, carbonaceous or woody. Hymenium types were classified as: pores, teeth, gills, daedaloid, apothecia, perithecia, cleistothecia, acosctromata, smooth, hysterothecia, gleba, or stings (Ulloa & Hanlin, 2012). For each species, we measured 40 spores; their size was estimated from mean length and breadth using the revolution ellipsoid, and spore shape was calculated as the length/width ratio (Gross, 1972).

The indices used to assess functional diversity in the four study sites were: functional richness (FRic), which represents the space filled by species in a convex hull volume of functional traits; functional divergence (FDig), representing the degree of niche differentiation as a result of functional trait dispersion in a community; and functional evenness (FEve), which indicates how homogeneously the abundance of characters is distributed in the trait space (Mason et al., 2005; Villeger, Mason & Mouillot, 2008; Mouchet et al., 2010). All the functional diversity indexes were calculated for the three trait groups with the FD package in R v. 3.4.2 (R Development Core Team, 2017).

We calculated the functional value index proposed by Avis et al. (2016) to assess the variation of macromycete ecological function along the urbanized gradient. For this, the recorded species were classified as ectomycorrhizal, saprotrophic, and parasitic fungi (based on authors knowledge and the specialized literature mentioned above) and a functional value was assigned to each functional guild (4, 3 and 1, respectively) based on the number of ecosystem services they provide (see Table 2.1 of Dighton, 2003; Tables S3a & S3b of Avis et al., 2016). The functional value was calculated for each of the 40 plots in the study area using the formula:

SiteSimple=[FEcM×ln(#speciesEcM+1]+[FSap×ln(#speciesSap+1]+[FPar×ln(#speciesPar+1]

where F is the functional value assigned to each guild, #species is the number of species, EcM is ectomycorrhizal fungi, Sap is saprotrophic fungi, and Par is parasitic fungi.

Statistical Analyses

Prior to statistical analyses, the assumption of normality was assessed performing Shapiro–Wilk tests (Shapiro & Wilk, 1965), and data were transformed when necessary, using Log, square-root, and reciprocal transformations. A two-sample Kolmogorov–Smirnov test was performed to define differences between patterns of species richness and diversity data obtained in the study sites; the null hypothesis for this test is that both data samples have identical distributions (Marsaglia, Tsang & Wang, 2003). The relationship between the number of species and the values of diversity was assessed with a linear regression analysis.

A regression tree analysis was performed to determine how the main explanatory variables affect the variation of macromycete species richness in the study area (De’ath & Fabricius, 2000). The correlation between species richness and explanatory variables was determined using the Spearman’s rho correlation coefficient. All the microclimatic, environmental and vegetation structure variables were included in both Spearman correlation and regression tree analyses.

We performed canonical correlation analyses (CCA) to understand macromycete distribution with respect to the set of environmental, microclimatic and urbanization variables. Forward selection was used for ranking explanatory variables in importance, and their statistical significance was tested using the Monte Carlo permutation test (Legendre & Legendre, 2012).

The relationship between every functional diversity index (FRic, FEve, FDig) and each urbanization variable, between every functional diversity index calculated by trait group (i.e., all traits, morphological traits, reproductive traits) and the level of urbanization along the study area, and between the functional value and macromycete species richness was analyzed by fitting linear and polynomial regressions using the Akaike information criterion (AIC) (Akaike, 1973) to select the best models.

We used one-way analyses of variance (ANOVA) to determine differences among study sites with regard to the functional value. To identify pairs of means that differed from each other, we used Tukey’s HSD tests with 95% confidence level.

All statistical analyses were performed in R Studio v. 3.4.2 (R Development Core Team, 2017).

Results

Macromycete species richness and diversity

A total of 223 individuals belonging to 134 species, 52 genera, 35 families, 16 orders, six classes and two phyla were recorded in the study area (File S1). The site with the highest urbanization level was Cerro el Fortín Ecological Reserve (Site 4), followed by Santo Domingo Tomaltepec (Site 3), Las Antenas (Site 2), and Benito Juarez National Park (Site 1). Benito Juárez National Park showed the highest number of species, diversity, and abundance (Site 1, Sobs = 77, 1D = 62.78, individuals = 116), followed by Las Antenas (Site 2, Sobs = 34, 1D = 29.11, individuals = 46), Santo Domingo Tomaltepec (Site 3, Sobs = 22, 1D = 19.7, individuals = 31), and Cerro el Fortín Ecological Reserve (Site 4, Sobs = 21, 1D = 18.57, individuals = 30). Patterns of species richness and diversity along the urbanization gradient did not differ significantly (Kolmogorov–Smirnov test, D = 0.5, P = 0.69), and were positively correlated (Linear regression, F = 34.19, r2 = 0.99, P < 0.0001). The species richness estimator Jacknife 2 indicated that the completeness of macromycete species inventories was 56.26 to 60.86%.

Shapiro–Wilk tests showed non-normal distributions of the data for soil and air humidity, tree density and height, litter depth and basal area. Thus, data were transformed for normality prior to analyses. The regression tree analysis (residual mean deviance = 7.168) indicated that the variation of macromycete richness in 22.5% of the samples (plots) of the study area was related to soil temperature <17.63 °C, while the number of species in 77.5% of the samples varied when soil temperature was >17.63 °C. Within this 77.5% of the samples, species richness in 32.25% of them was influenced by soil temperatures <19.58 °C, and in 65.27% of them it varied due to soil temperature >19.58 °C. Similarly, from this 65.27% of samples, species richness within 28.5% of them was influenced by air humidity below 54.47%, whereas species richness in 71.5% of these samples varied with higher air humidity (File S2).

Spearman correlation analyses indicated that all microclimatic variables were significantly related to macromycete species richness within the study area. Both air and soil temperature were negatively correlated with the number of species (rho = −0.56, P = 0.0001; rho = −0.73, P = 7.8E−08, respectively), whereas soil and air humidity were positively correlated (rho = 0.64, P = 7.1E−06; rho = 0.35, P = 0.02, respectively). With regard to environmental variables, slope and litter depth were positively correlated with the number of macrofungal species (rho = 0.31, P = 0.04; rho = 0.57, P = 9.32E−05, respectively), while stoniness and soil cover showed a negative correlation with richness (rho = −0.43, P = 0.005; rho = −0.37, P = 0.01, respectively). Mean tree height was the only vegetation structure variable significantly correlated with macromycete richness (rho = 0.39, P = 0.01) (Table 1).

Table 1 Spearman’s rho coefficients (ρ). Correlation between macromycete species richness and explanatory variables.

Variable	ρ	p-value	
Slope*	0.317	0.045	
Aspect	0.101	0.532	
Stoniness cover**	−0.433	0.005	
Soil cover*	−0.375	0.016	
Litter depth***	0.578	<0.0001	
Air temperature***	−0.569	0.0001	
Soil temperature***	−0.732	<0.0001	
Air humidity*	0.359	0.022	
Soil humidity*	0.644	<0.0001	
Tree density	−0.147	0.363	
Mean tree height*	0.390	0.012	
Basal area	0.277	0.082	
Notes:

* p < 0.05.

** p < 0.01.

*** p < 0.001.

Macromycete species composition and distribution

The Chao-Jaccard similarity index showed a high turnover of species composition among study sites. Species differed from 75.3 to 94.4%; the highest similarity in species composition occurred between Sites 3 and 4 (24.9%), and the lowest was observed between Sites 1 and 3 (5.6%) (File S3).

The CCA for microclimatic/environmental variables included air and soil humidity, air and soil temperature, slope, aspect, soil and stoniness cover, litter depth, tree density, basal area, and mean tree height, and it was performed for the 136 macromycete species recorded in the study area. Axis 1 (eigenvalue = 0.88) and axis 2 (eigenvalue = 0.84) accounted for 10% and 9% of the relationship between species distribution and explanatory variables, respectively (Monte Carlo test, axis 1, F = 1.4, P = 0.007; axis 2, F = 1.34, P = 0.024). CCA results showed a clear separation between Site 1 and the other sites along axis 1, and several species were strongly associated with this forest where air and soil humidity, slope, and mean tree height are the most relevant variables. Site 2 is distinguished from Site 3 along axis 2, and the type of soil cover and aspect were the variables most related to Site 2. Site 4 did not show a clear trend along axis 2 (Fig. 1).

Figure 1 CCA for the macromycete species recorded in the four study sites.

Vectors are microclimatic, environmental, and vegetation structure variables: aspect, soil cover (SoilC), air humidity (airH), soil humidity (soilH), air temperature (airT), soil temperature (soilT), slope, aspect, stoniness cover (StoneC), mean tree height (treeH), tree density (treeD), and basal area (BA).

The CCA for urbanization variables included the extent of crop and built areas, population density, extent of introduced vegetation areas, street and road length, and 136 macromycete species, but the model retained only three variables. Axis 1 (eigenvalue = 0.86) and axis 2 (eigenvalue = 0.77) accounted for 37% and 33% of the relationship between species distribution and urbanization variables, respectively (Monte Carlo test, axis 1, F = 1.37, P = 0.001; axis 2, F = 1.22, P = 0.003). Site 1 was clearly separated from Sites 2, 3 and 4 along axis 1 (Fig. 2). The distribution of Site 2 was defined by the extent of crop area along axis 2. Sites 3 and 4 were not distributed apart from each other and were strongly related to street/road length and extent of built area (Fig. 2).

Figure 2 CCA for macromycete species recorded in the four study sites.

Vectors correspond to urban variables: extent of crop areas (Crop), extent of built areas (Built), and length of roads and streets (stLength).

Functional diversity and functional value in macromycete communities

The functional richness index (FRic) was the highest in Site 1 (10.66), followed by Site 2 (2.32), Site 3 (0.04) and Site 4 (0.01). The Spearman correlation coefficient indicated a positive relation between species richness and FRic (rho = 0.99, P = 5.29-7), and the patterns of these metrics were similar throughout the study area (Kolmogorov–Smirnov, P > 0.05). Functional divergence (FDig) showed its highest value in Site 4 (0.88), followed by Site 2 (0.77), Site 3 (0.76) and Site 1 (0.72). The Spearman correlation showed a negative relationship between species richness and FDig (rho = −0.78, P = 0.01). Functional evenness (FEve) did not show a significant trend along the urbanization gradient. It was the highest in Site 2 (0.63), followed by Site 1 (0.61), Site 4 (0.57) and Site 3 (0.54).

The AIC designated linear regression as the best fit model to evaluate the relationship between the level of urbanization along the study area and the functional diversity values obtained for the different trait groups (i.e., all the traits, morphological traits, and reproductive traits). FRic was negatively related to urbanization level when calculated for both all the traits and morphological traits (r2 = 0.5, F = 6.05, P = 0.49; r2 = 0.68, F = 13, P = 0.01, respectively; Figs. 3A1, 3B1). FDig for all the traits showed a positive relation with urbanization gradient (r2 = 0.75, F = 18.2, P = 0.005; Fig. 3A2). The rest of these results for FRic, FDig and FEve were not statistically significant (P > 0.05) (Fig. 3).

Figure 3 Linear regression analysis for functional richness (FRic), functional divergence (FDig), and functional evenness (FEve) along the urbanization gradient.

Analyses were performed using three trait groups: all traits, morphological traits, and reproductive traits.

Linear regression was the best fit model designated by AIC to evaluate the relationship between the obtained FDig values and every urbanization variable, whereas linear and polynomial regressions were designed for FRic and FEve. FDig was the only index significantly related to urbanization variables; the analysis indicated a positive relation with the extent of built areas (r2 = 0.85, F = 36.63, P = 0.0008; Fig. 4C1), street/road length (r2 = 0.83, F = 31.39, P = 0.001; Fig. 4C3), and population density (r2 = 0.83, F = 31.12, P = 0.001; Fig. 4C4). FRic and FEve were not significantly related to the urbanization variables (P > 0.05) (Figs. 4A, 4B).

Figure 4 Linear regression analysis between functional richness (FRic), functional divergence (FDig), functional evenness (FEve), and urbanization variables.

The recorded macromycetes included 75 species of ectomycorrhizas, 53 species of saprotrophs, and 6 species of parasites (File S1). The functional value had an average of 8.74 and a range of 3.9–15.27, being markedly higher in the less urbanized site (Site 1) (File S4). The ANOVA for the functional value indicated statistical differences between sites (DF = 3, F = 16.53, P < 0.0001), and the Tukey’s HSD test revealed that Site 1 differs significantly from Sites 2, 3 and 4 (P < 0.001), but Sites 2, 3 and 4 do not differ from each other (P > 0.05). Polynomial regression was the best fit model designed by AIC to evaluate the relation between functional value and macromycete species richness, and the analysis showed a positive relation (r2 = 0.84, F = 201.5, P < 0.0001; Fig. 5).

Figure 5 Polynomial regression analysis between the functional-value index and macromycete species richness.

Dots represent every plot in the study area.

Discussion

This is the first study to analyze the functional diversity of macromycete communities in an urban ecosystem in Mexico. Our results indicated a marked decline of macromycete richness, diversity and functional diversity with greater urbanization, and are consistent with studies conducted along urbanization gradients showing that species richness drastically decreased from the outskirts towards the urban core (Niemelä, 1999; Marzluff & Ewing, 2001; McKinney, 2006). The decline of species richness with higher urbanization levels can be explained by the modification of biological interactions and habitat transformation. Macromycetes require interactions such as mycorrhizal-plant host and plant remains as substrates, thus changes in the structure of tree communities resulting from urbanization can influence macromycete richness by providing different habitats and affecting the quality and quantity of the available resources (Ferrer & Gilbert, 2003; Richard et al., 2004; Brown, Bhagwat & Watkinson, 2006; Newbound, Mccarthy & Lebel, 2010; Zhang et al., 2010). Besides, it is broadly known that habitat transformation can strongly affect microclimate conditions and diminish the number of fungi species (Lodge et al., 2004; Gómez-Hernández et al., 2012).

All the microclimatic variables analyzed in our study were related to species richness along the study area, but soil temperature and humidity showed the highest correlation (Table 1), and their variation was concordant with the variation of diversity, number of species and number of individuals among sites. In accordance with our results, other studies exploring macrofungal richness and abundance at the local scale in both tropical and temperate regions have shown them to be strongly related with environmental/microclimatic factors, mainly humidity and temperature (O’Dell, Ammirati & Schreiner, 1999; Brown, Bhagwat & Watkinson, 2006; Durall et al., 2006; Gómez-Hernández & Williams-Linera, 2011; Caiafa et al., 2017). Even though humidity and temperature are key factors related with macromycete richness, distribution, and fruit body production worldwide, this relationship may diverge depending on regional differences (Boddy et al., 2013). For example, in arctic-boreal regions, local increases of temperature can foster fungi development and production, whereas the high-water content in these regions can negatively affect macromycetes (Boddy et al., 2013; Blanchette et al., 2016). In tropical regions, by contrast, studies that correspond with our results have shown that soil and air temperature are negatively related to macromycete richness and abundance, whereas soil and air humidity are positively related (Lodge & Cantrell, 1995; Brown, Bhagwat & Watkinson, 2006; Caiafa et al., 2017; Ruiz-Almenara, Gándara & Gómez-Hernández, 2019). In our study area, Site 1 recorded the maximum diversity and number of macromycete species and individuals, followed by Site 2, Site 3, and Site 4. Correspondingly, Site 1 presented the maximum soil humidity and the lowest soil temperature (29.5 % and 16.9 °C, respectively), followed by Site 2 (8.4% and 19.4 °C), Site 3 (8.2% and 20.1 °C), and Site 4 (6.7% and 20.4 °C). However, changes in temperature and humidity affecting macrofungal communities along geographic or environmental gradients have been suggested to be a function of the varying vegetation structure (e.g., canopy openness, basal area, tree height, tree density) and environmental factors (e.g., slope, leaf litter depth, aspect, soil cover) (Brown, Bhagwat & Watkinson, 2006; Cavender-Bares et al., 2009; Zhang et al., 2010; Singha et al., 2017; Gómez-Hernández, Ramírez-Antonio & Gándara, 2019).

Our results showed that the variation of both species richness and distribution were related to vegetation structure and environmental variables influencing microclimatic conditions (Table 1; Fig. 1). Vegetation structure and organic matter produced by trees are among the main factors affecting macromycete communities in the forests due to the heterogeneity of habitats, resources, and resource quality they provide. Additionally, the water-holding capacity of dead organic matter on the soil, such as leaf litter, promotes soil water content and avoids the increase of temperature and loss of soil humidity, which are highly relevant for macrofungal growth (Villeneuve, Grandtner & Fortin, 1989; Ferris, Peace & Newton, 2000; Egli et al., 2010; Gómez-Hernández & Williams-Linera, 2011). Moreover, factors linked to the topography of a landscape (e.g., soil surface unevenness, slope, aspect) can influence water drainage, evaporation rate, wind exposure, and, in turn, soil and air temperature and humidity, affecting fruit body production, and species richness and distribution (Nantel & Neumann, 1992; Rubino & McCarthy, 2003; Gómez-Hernández et al., 2012; Caiafa et al., 2017).

Similarly, urban factors appear to affect macrofungal communities along our study area (Fig. 2). Macromycete species throughout the urbanization gradient are impacted mainly by the length of streets/roads and the extent of built areas and, to a lesser extent, by the extent of crop areas. Studies have suggested that impervious surfaces through urban areas can lead to poor water absorption capacity by the soil, and drastically affect environmental temperature and humidity (Newbound, Mccarthy & Lebel, 2010). In addition, temperature in urban landscapes can increase owing to the heat island effect, which may be a consequence of the urban surface capacity to absorb solar radiation, and the heterogeneous distribution of vegetation in urbanized areas (Arnfield, 2003; Boone & Fragkias, 2012), thus resulting in a decrease in macromycete richness and fruit body production, and changes in species composition (Ferris, Peace & Newton, 2000; Rubino & McCarthy, 2003; MacGregor-Fors et al., 2015).

Within our studied area, functional richness (FRic) significantly decreased with urbanization level (Fig. 3A1), and this corresponded with the observed trend in macromycete species richness. The relationship between FRic and the number of species in biotic communities may be explained, in part, because the number of species reflects the number of conserved traits through the taxonomic lineages (Caiafa et al., 2017). This functional component of diversity gives species richness a main role in the stability and functioning of ecosystems, influencing processes such as productivity or resilience (Diaz et al., 2006; Mouchet et al., 2010; Sol et al., 2020). Since FRic is a measure of richness indicating the amount of niche space occupied by the species in a community, low values of FRic in urbanized sites of our study area suggest that fewer available resources within these communities are being used. This may be explained because the loss of macromycete species caused by urbanization can lead to communities with a narrow range of functional traits (which are directly linked to taxonomic differences), and assemblages of species unable to take advantage of potentially available resources (Mason et al., 2005; Caiafa et al., 2017). FRic also decreased throughout the urbanization gradient in our study area when calculated with morphological traits (Fig. 3B1).

Our findings showed that functional divergence (FDig) significantly increased with urbanization level when analyzed using all the traits (Fig. 3A2) and was positively related to urbanization variables (Figs. 4C1, 3, 4). FDig measures the degree of niche differentiation based on the distribution of the total abundance in the niche space supported by the species with the most extreme functional traits (Mason et al., 2005; Mouillot et al., 2013). The increasing FDig with urbanization level indicates a high degree of niche differentiation among macromycete species within communities in urbanized areas. This means that species with higher abundance are functionally dissimilar, thus they occur at the extremes of the niche space (Mason et al., 2005; Mouchet et al., 2010). Since most of the resources are being used by species with the most extreme functional traits, the results indicated that the scarcely occupied niche space and under-utilized resources in urban communities, suggested by FRic, are close to the center of the functional trait range (Villeger, Mason & Mouillot, 2008). Functional evenness (FEve) did not show significant variation along the urbanization gradient (Figs. 3A3, 3B3, 3C3). FEve measures the regularity in the distribution of species abundance along the functional space for a given community, and our results suggest that in spite of urbanization having caused a reduction in the niche space occupied by macromycete species (indicated by FRic) and a high niche differentiation (indicated by FDig), there is not much variation in the distribution of abundances within the occupied parts of the niche space (Mason et al., 2005; Mouillot et al., 2013).

The morphology-related traits that we selected to calculate the functional diversity indexes are “response traits” (i.e., they can affect organism’s performance under different environmental conditions), and are associated to macromycete dispersion, establishment, uptake of new resources, and reproduction. The fruit body texture relates to timing of spore production due to its link to fruit body longevity, which can vary from few days to several months; hymenium type is involved in how spores are released; pileus diameter is related to hymenial area and hence to spore production; and spore size and shape are related to dispersal fitness, dormancy, and germinability (Moore et al., 2008; Norden et al., 2013; Bässler et al., 2015). The functional diversity indexes indicated that some sections of the niche space are unoccupied and available resources underutilized in the most urbanized sites, suggesting that the microclimatic/environmental conditions derived from urbanization are reducing the morphology-related trait diversity, and hence macromycete species performance. Since macromycetes play a main role in terrestrial ecosystems, their limited capacity to disperse, establish and reproduce in the niche space, and use available resources can affect plants access to nutrients, rates of organic matter decomposition, and regulation of organismal populations (Deacon, 2006, Egli, 2011, Zanne et al., 2020), which could affect the ecosystem functioning and productivity (Petchey, 2003; McKinney, 2006).

Studies have found that changes in the structure of tree communities can impair macromycete functional guilds mainly due to the physicochemical attributes of the soil are altered as well as the quality and quantity of available organic matter (Dighton & Mason, 1985; Fernández-Toirán, Ágreda & Olano, 2006; Gómez-Hernández, Ramírez-Antonio & Gándara, 2019). The functional value metric showed that Site 1 (National Park) differs widely from the other sites regarding macromycete functional attributes, whereas Sites 2, 3 (rural areas) and 4 (city) do not differ from each other. Since this metric characterizes functional guilds and their prevalence in a site (Avis et al., 2016), the results suggest that the impact of urbanization on the ecological function of macromycetes can become equally drastic in both the urban core and the rural hinterlands. Despite most studies evaluating patterns of macromycete diversity report an increasing diversity from disturbed to conserved areas, evidence indicates that this trend can differ when functional guilds are evaluated (Fernández-Toirán, Ágreda & Olano, 2006; Vásquez et al., 2011), which means that species richness and functional guild approaches can provide different information. Our results showed a positive correlation between functional value and species richness, and a high turnover of species composition among sites, making this information of interest to infer about the variation of the ecological functioning of macromycetes along the study area. However, it has been suggested that functional value is a more informative metric than species richness when assessing fungal ecological function, and offers a better predictive framework than species composition for understanding fungal function across systems with few species in common, which can be explained because species diversity and distribution at local scale are determined mainly by niche differences and competition (Kraft & Ackerly, 2010; Talbot et al., 2014; Avis et al., 2016).

The present study included only one season of data, which could be a limitation. However, sampling was carried out on several dates considering fungal phenology and recorded more than 50% of the species. Since the sampling procedure was the same in the four study sites, the obtained data can be compared between sites and get suitable results to understand how diversity varies along the studied area. Owing to the lack of appropriate lab equipment and experience analyzing atmospheric and soil pollutants, our study did not evaluate their effect on macromycete communities. However, evidence indicates they are urbanization factors that negatively impact fungal communities, altering their performance within urban environments (Grimm et al., 2008; Newbound, Mccarthy & Lebel, 2010). This evidence should prompt future macromycete studies to include this kind of urban variable.

Conclusions

This study suggests that urbanization is negatively affecting macromycete diversity from the less-urbanized sites to the core of the urban area, mainly due to the high temperature and low humidity resulting from intense land-use and forest disturbance. The loss of species has led to a decrease in functional diversity within communities, resulting in available resources scarcely being used by species and niche sections unoccupied. Moreover, our results indicate that macrofungal communities within the less-urbanized site present greater niche complementarity, higher variation of functional traits, and greater functional guild performance. Analyzing functional diversity using different trait groups (i.e., all traits, morphological traits, and reproductive traits) and functional guilds (e.g., mycorrhizas, saprotrophs, and pathogens) allowed us to obtain more robust results and generate relevant information. In this respect, diversity metrics based on functional traits constitute a powerful tool to assess macromycete communities along environmental and geographic gradients (Avis et al., 2016). However, it would be interesting for studies of functional diversity to integrate metrics based on phylogenetic information in order to elucidate the evolutionary history among species and understand how urbanization affects the phylogenetic structure of communities, which can be highly valuable for management and conservation planning efforts (Branco & Ree, 2010; Gómez-Hernández et al., 2016).

Supplemental Information

Supplemental Information 1 Macromycete species, Families, Orders, Classes and Phyla recorded within the studied area at Southwest, Mexico.

“sp., sp. 1, sp. 2, sp. 3…” indicate species unidentified (morphospecies). The functional guilds (FG) are ectomycorrhizas (E), saprotrophs (S), and parasites (P).

Click here for additional data file.

Supplemental Information 2 Regression tree for macromycete species richness.

Each partition shows the explanatory variable and threshold at which the partition was made. The average value of the variable effect is indicated at the tips and nodes. Variables considered are soil temperature (SoilT) and air humidity (AirH).

Click here for additional data file.

Supplemental Information 3 Chao-Jaccard similarity index.

Matrix indicating similarity between study sites based on species composition.

Click here for additional data file.

Supplemental Information 4 Functional-value index (FV), macromycete species richness (MacroM), and number of ectomycorrhizal (EcM), saprotrophic (Sap), and pathogenic (Pat) fungal species observed in each of the 10 plots of the four study sites.

Click here for additional data file.

Supplemental Information 5 Macromycete species/morphospecies and abundances recorded in each plot of the four study within the study area in Southwest Mexico.

Click here for additional data file.

Supplemental Information 6 Microclimatic, environmental, and vegetation structure variables recorded in each plot of every study site in Southwest Mexico.

Click here for additional data file.

We thank Dra. Sandra Smith Aguilar for helpful suggestions and comments on earlier versions of this manuscript; Dr. Peter Avis and one anonymous reviewer for important and useful comments that greatly improved the manuscript.

Additional Information and Declarations

Competing Interests

Author Contributions

Field Study Permissions

Data Availability

The authors declare that they have no competing interests.

Marko Gómez-Hernández conceived and designed the experiments, performed the experiments, analyzed the data, prepared figures and/or tables, authored or reviewed drafts of the paper, field work, and approved the final draft.

Emily Avendaño-Villegas conceived and designed the experiments, performed the experiments, analyzed the data, prepared figures and/or tables, field work and specimens identification, and approved the final draft.

María Toledo-Garibaldi performed the experiments, analyzed the data, authored or reviewed drafts of the paper, and approved the final draft.

Etelvina Gándara conceived and designed the experiments, prepared figures and/or tables, authored or reviewed drafts of the paper, specimens identification, and approved the final draft.

The following information was supplied relating to field study approvals (i.e., approving body and any reference numbers):

Authorizations to carry out field work in San Pablo Etla, San Perdo Ixtlahuaca and Santo Domingo Tomaltepec were given verbally, the local authorities were Oscar Zárate Juárez, Ernesto Gabriel García Ramos and Celestino Soto Martínez, respectively.

The following information was supplied regarding data availability:

The raw measurements are available in the Supplemental Files.

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
