# Peer review of "Impact of urbanization on functional diversity in macromycete communities along an urban ecosystem in Southwest Mexico"

_PeerJ, doi:10.7717/peerj.12191_

## Round 0.1 · original submission · Major Revisions

Both reviewers have indicated that your study has merit, but that there are many revisions required to improve the readability and make the study more accessible and relevant for a wider audience. Reviewer 1 has suggested some possible additional lines of investigation in terms of focusing on particular taxa groups which could also improve the quality of your findings. In particular, the discussion section could be improved, and likely shortened as per suggestions of Reviewer 2.

·

Basic reporting

Some areas in need of improved English. Some additional references encouraged. Mostly professional structure, figures and tables, but see comments. Needs a clearly stated hypothesis in the Introduction.

Experimental design

Research seems to be original and a question/objective defined that is relevant and meaningful. Gap addressed could be made more universal. Investigation appears rigorous though methods need improved clarity and additional information.

Validity of the findings

Replication in terms of multiple urbanization gradients tested would be an improvement. As is, the study only examines one gradient. Each location along the single gradient has multiple plots (i.e. pseudureplicates). Underlying data not available, unsure of where they can be found. Conclusions are mostly clear and linked to original questions.

Additional comments

The authors characterize the “macromycete” fungal communities across an urbanization gradient. The key results show that soil temperature and air humidity are driving factors in species richness and that the urbanization gradient (which combines street/road amounts, building density) is negative for functional richness but positive for functional divergence (the latter I am still not sure I fully understand). Overall, this is a reasonable study but I am unsure of how interesting it might be beyond a small set of readers. The case could be made more convincingly that this is a global and not a solely Mexican pattern. Study needs revision, many small items to attend to and several larger considerations.

One area to expand and reconsider: Look at the patterns of mycorrhizal, saprotroph and parasite groups separately...do they do different things across the gradient? See the Gomez-Hernandez et al 2019 Fungal Ecology paper cited

Figure 4 and 5 (maybe even Fig 1) and Table 2 could go to the supplement without harm.

Line (number given followed by “:” and suggested correction or comment) by line comments:

28: mutualists
28: pathogens, mutualists and decomposers…
32: I think the functional aspect is more about the specific ways in which the organisms processes energy and matter and less to do with evolution.
35: in order to infer
41: with increased level of
42: sites which appear to be
51: But in line 44-45 it says FDiv increased with urbanization. This will be confusing until readers further get into the paper. It probably should be stated as FRic not “functional diversity”
75: Do not use the word “Nowadays” here and elsewhere. Can start the sentence at the following word.
82: Consider changing "Macromycetes" to "Fungi" at least here in the introduction. Then, in the methods, clearly state that this study only focuses on "macromycetes" and then explain what criteria the study used to determine and survey only macromycetes.
98: Important instead of relevant
114: See Avis et al Restoration Ecology https://doi.org/10.1111/rec.12397 for an example of using functional diversity in the context of ecological restoration including some sites of urban settings.
120: It would be preferred if the authors propose specific hypotheses about the relationship between urban gradients and macromycetes.
156-158: The sentence seems more like it belongs in the Results section.
162: Was there a size or diameter minimum that was considered to be "macromycete" at least according to this study? this needs to be clarified so that we understand how only "macros" were considered. Where is the boundary between macro and micro in this study?
163: How or why was this determined? At a minimum, state references that support this contention and point out limitations of this choice. There is very good evidence that some "macromycete" individuals are extremely large while others are quite small -- and I would surmise that population genetic tests were not conducted on all of these species to determine individual genet size. So, this study's criteria, as written, has not accounted for this. Since this is an unknown, it would be better to state this as “observations” and not “individuals”. Then in the methods describing the diversity metrics make it very clear that this assumption is not met and that the study uses “observations” instead of true measures of individuals.
177: What’s the rationale for three groups? What were “all traits”? Why not just look at trophic group?
187: For each functional index, explain how the index was calculated. For example, was FRic the total number of functional types (including all three trait groups)? How did this handle repeats of a trait group? These need to be clearly defined FDiv and FEve as well.
188: remove “convex hull”
197: Were data normal and identically distributed? In the results ,there is no mention of this and/or any subsequent transformations conducted to deal with data and residual issues.
210: What is “nutritional”? Trophic groups? Not mentioned or described previously.
215: Better stated as "collections" or "observations" rather than individuals.
224: Given that just more than half of the communities should we expect the answer to change? This issue is not at all dealt with in the Discussion section and should be. Or, can the authors survey for another year? If they could, this study would be much stronger.
243: species, no “of”
244: Jaccard not mentioned in Metods
245: similarity
277: Can a table comparing the different models used and their respective AIC be shown? This will help the reader see the different models considered.
278: indices
295: What is “true”? Explain.
323: should read "...have shown that..."
324: The global relationship is probably hump-shaped but under tropical conditions, such fungal communities are at the downward slope of the hump (and conversely, in boreal-arctic regions the relationship is at the start, the upslope of the hump). This might be worth clarifying if the readers are globally focused. Urbanization happens globally.
350: should read "...appear to affect..."
386: Since the text that follows is a list that explains the statement prior, use a ":" instead of ";" Alternatively, and my preference, would be to just make these all shorter sentences.
399: stop the sentence after "...2013). The increasing..."
404: more urban communities
405: Not sure I understand this point. Could it be elaborated? How does this show the center of the trait range?
420: Should say "...study suggests that urbanization..." Correlation is not causation and this study just looks at the relationship. It does not experimentally test the effect of urbanization which is what would be needed to state it in the way the authors wrote this.
430: Again, worth referencing Avis et al Restoration Ecology 2016
433: Reference here e.g. Peay et al New Phytologist (2010) 185: 529–542 and Branco and Ree (2010) PlosOne https://doi.org/10.1371/journal.pone.0011757
504: Mycorrhizal

Reviewer 2 ·

Basic reporting

no comment

Experimental design

no comment

Validity of the findings

no comment

Additional comments

To the Authors,

I have now reviewed manuscript ID: #59452, entitled ‘Impact of urbanization on functional diversity within macromycete communities along an urban ecosystem in Southwest Mexico’. This study sought to determine the effects of an urbanization gradient on macromycete communities within an urban ecosystem of Southwest Mexico. The authors have measured various macromycete functional and trait diversity metrics, along with community composition, and explanatory variables of the biotic metrics above.

Major concerns:
The overall writing style, tone, language, and clarity need improving to properly reflect the study conducted. This will improve the overall comprehension as a reader of the manuscript. Various parts of the writing style throughout the manuscript need to be toned down to keep within the aim, scope, and limitations of the study. I suggest the authors consult with an editing service or colleague to help proofread and revise this manuscript. Methods need to be more extensive to ensure the repeatability of the study; some parts remain unclear; a few questions pertaining to methods are asked and provided in the general comments section below.

Weaknesses: Writing style needs major revision to convey the aim, scope, limitations and justifications in a comprehensible manner to improve overall clarity. More explanation should be provided in the Introduction to help guide the reader along as to the differences between ‘macromycetes’ and broadly based fungal groups, and more information about the important roles of fungi and their intimate connection with their surround environment, as PeerJ is not a microbial ecology-based journal.

Strengths: Important to understand fungal community patterns in urban ecosystems, and even more so for sub-tropical/tropical ecosystems. Hand-sampling fungi is not easy; consider DNA metabarcoding approach in the future, to assess fungal community patterns from environmental samples collected in this region—food for thought.

General comments and concerns:

Title
The title is a bit unclear and suggest revising to improve comprehension and clarity in order to accurately convey the study being presented. For example, replacing “macromycete communities” with “fungal communities”—this may help engage the reader more. Moreover, these fungal communities were examined along an urbanization gradient within an urban ecosystem of Southwest Mexico; would suggest rephrasing the title to

Abstract
The Abstract could be shortened to keep within the aims, scope, and limitations of the study (e.g., this is a case study; please tone down language and writing style). Moreover, is there a justification in using the term ‘macromycetes’ as opposed to ‘fungi’? The difference between macromyete-fungi and fungi, should be clear to the reader in the Abstract, as PeerJ is not a microbial ecology-based journal. Abstract needs to be revised such that the reader is guided along a story as to what the problem is with urbanization and macromycetes; knowledge gap for macromycetes in this area; why are macromycetes important.

Please keep any interpretations to the Discussion section.

Ln 25-27: This sentence lacks context and should be revised or deleted. For example, “abundance of individuals” does not provide the reader as to what type of individual is being referred to (i.e., this study only examines macromycetes); likewise, “loss of species which play different roles in ecological communities” is vague and shrouds the overall cohesiveness and clarity of the message being conveyed.

Ln 27-30: This sentence lacks context and should also be reorganized to improve comprehension and clarity.

Ln 30: Please specify the species organisms you are referring to (i.e., macromycetes); and be consistent throughout the manuscript to remind the reader.

Ln 30-32: What is meant by “they incorporate an evolutionary background”? This sentence should be revised; unclear.

Ln 35: “to infer the effect of urbanization on macromycete communities” can be omitted. Would indicate/incorporate this earlier in the sentence (i.e., aim of the study was to assess patterns of macromycete species richness, functional diversity, and community composition along an urbanization gradient in an urban ecosystem of SW Mexico).

Ln 36-38: This sentence can be shortened and/or condensed with previous sentence (not necessary to put in the dimensions of the plots in the abstract).

Ln 43-44: Interpretations should be kept to the Discussion section.

Ln 44-46: This sentence would be more appropriate as Discussion material, not Abstract material.

Ln 46-48: Keep interpretations to Discussion section.

Ln 50-52: This sentence is speculative as “ecosystem productivity” and “stability” were not measured. Please refrain from using this type of language throughout the manuscript.

Ln 48-52: These sentences need to be revised to keep within the aims and scope of the study, and what was actually measured and what the data and results show.


Introduction

Justification could be built upon by focusing on knowledge gaps in the research and the critical roles that fungal communities play in not only non-urban ecosystems, but urban ecosystems as well.

Can condense paragraphs 1 and 2 to shorten the introduction and expand on various topics to make the introduction more relevant, cohesive, and comprehensible for readers that are not familiar with microbial groups such as fungi (e.g., Macromycetes).

Ln 59-60: How many decades? Also, this sentence could use an in-text citation.

Ln 70-72: Which research? Please reference. Moreover, this sentence needs more context as this study is only focusing on fungal groups, and in saying “exploring biotic communities” does not provide the reader with context to comprehend more specifically the message being conveyed. Same with “declines in species richness” and “abundance of individuals”. The end of this sentence—“there is a decreasing continuum of human influence from city centres out toward wild areas” is a bit vague and seems to be a bit of a non-sequitur. Please consider revising this sentence.

Ln 72-74: It is not clear which kind of biotic communities the authors are referring to—in addition, is this the case for all biotic communities? Please be consistent in using the organismal group being studied (i.e., macromycetes), if this is true for this group.

Ln 77: Please cite this reference here, then provide the results of this study in the following sentence starting with “The results showed…”

Ln 77-81: This sentence is a bit lengthy, consider revising.

Ln 82: This sentence needs a reference.

Ln 82-88: Unclear the difference between Macromycetes and fungi—this needs to be explained more in the Introduction for those readers that are unfamiliar with fungi.

Ln 85-90: Do Macromycetes fall within these categories of fungi? (i.e., saprotrophic, wood-decaying, and mycorrhizal fungi). This is unclear in these sentences.

Ln 96-99: This sentence could be rephrased to provide more context (i.e., evaluating variation in Macromycete species diversity and community distribution patterns…..).

Ln 99-101: What is meant by this? What kind of species? Do other species have different responses? This sentence is vague and lacks context that is pertinent to the study—should be revised.

Ln 101: Would be a good idea to provide an example of a specific Macromycete functional trait.

Ln 117-120: The aims and research questions could be made more clear.

Methods

Ln 129-134: Is there evidence to suggest elevation could be a contributing factor?

Ln 132-134: What is the justification for these dimensions? (i.e., why 2 large plots for each site, and 5 within each plot?)

Ln 136-144: These methods should be made more clear such that the study is repeatable. How were these explanatory variables measured? Proper instrumentation should be given as well where appropriate.

Ln 145-158: Is this a standard method in identifying urbanization gradients? References would be useful here.

Ln 160-167: This section needs a more extensive description of sampling methods and applicable references (i.e., are these standard procedures for sampling macromycetes?; Were any sterile techniques applied?; Are sterile techniques necessary?; Is there a standard procedure or reference in which to identify micro- and macro-morphological characteristics?).

Ln 172: Change “was” to “were”.

Ln 173: Justification for using Jaccard similarity when the data has been presented as abundance, as opposed to presence/absence?

Ln 172-174: These methodological descriptions need to be more extensive.

Ln 176-178: Reference?

Ln 178-181: Reference?

Ln 176-193: The functional and trait diversity methods need to be made clearer here and referenced appropriately.

Ln 195-198: These sentences are unclear—please consider revising.

Ln 195-203: Statistical analyses need to be made clearer and explicit, as well as properly referenced analyses. Moreover, were the biotic data standardized prior to analyses? Were the abiotic data normalized prior to analyses? This is not clear.

Ln 199-211: These sentences need to be revised and made more extensive to guide the reader along as to what analyses were used for which specific research questions.

Results

Ln 223-224: What is meant by “complete”? This is unclear.

Ln 225-228: Please revise these sentences—not comprehensible.

Ln 229: “Within the latter group of sample”….—easier to refer to the samples as opposed to this language, to improve readability.

Ln 229-232: These sentences are unclear—please revise.

Ln 233-242: Please consider providing these data results in a table of the manuscript.

Ln 233-242: Please be consistent in p-value significant figures.

Ln 243: Typos

Ln 244-246: This is unclear.

Ln 251: The Monte Carlo methods are not described in this manuscript.

Ln 277-278: This is not clear

Ln 285-286: What does this mean?

Discussion
The Discussion section is dense with in-text citations and suggest removing where appropriate. This section also needs to be shortened in length. Moreover, some of the information provided in this section may be more fitting as Introduction material. Some pieces of information are difficult to interpret, and it is unclear in various areas throughout this section as to how the authors have arrived at their conclusions, supported by their data/results and why; there does not seem to be a clear link of their results and the discussion material provided. Please omit from using strong speculative language when discussing the interpretations of the results, and please keep within the aims, scope, and limitations of the study.

Ln 301-308: More appropriate as Introduction material.

Ln 301-307: Lengthy sentence—please consider shortening.

Ln 309-314: More appropriate as Introduction material.

Ln 315-318: Please interpret and discuss the results; the discussion section should not be a repeat of the results.

Ln 318-321: This sentence is unclear; how is this related to an urbanization gradient? Please consider revising.

Ln 327-336: These sentences can be condensed.

Ln 361-366: How is this related to the results?

Ln 369-371: The message being conveyed in this sentence is not clear. Please revise.

Ln 378: The amount of niche space is not described well in the introduction.

Ln 378-383: These sentences are unclear, and it is not clear to the reader how the authors arrived at this conclusion. Please revise.

Ln 385-390: Please shortened—too long.

Ln 394: This study did not test for ecosystem productivity; please keep within the aim and scope of study.

Ln 401-403: This is not explained well—please revise.

---

## Round 0.2 · Minor Revisions

Your manuscript has been deemed acceptable by one reviewer, but the other reviewer still has serious concerns. I would ask you to consider his comments regarding the use of 'functional' and 'effect' traits, and respond to them. There is a suggestion that you may wish to carry out some further analyses, and I would ask you to address this in your response whether or not you choose to do this.

·

Basic reporting

Yes, sufficient.

Experimental design

There remains concern in how the interpretations of "functional" are portrayed. See below.

Validity of the findings

See "additional comments".

Additional comments

This manuscript is improved and several areas are much better than the previous version. I appreciate that the authors have considered the previous comments seriously and also that they have pushed me to study deeper the current state of functional trait ecology. However, several key areas still need to be addressed in order to make this manuscript acceptable for publication.

The interpretation provided for “functional” and the extent to which the authors extend interpretations and discussions beyond the limits of the evidence provided remain a serious point of concern. In essence, they have measured five “morphology-related” (line 427) traits of the macrofungi surveyed in the forests of different sites along the urbanization gradient. From these data, they have extrapolated in the discussion about how the responses measured illustrate a “reduction in ecosystem productivity” (line 429) and other processes.

Given only the morphological evidence, it is not at all clear how these could be connected to ecosystem productivity or other processes. The five traits selected have no clear (at least as defined by the authors to this point) and direct relationship to ecosystem productivity, function or service -- unless they are presuming that pileus size and spore production are elements of ecosystem productivity substantial enough to compare to primary productivity by autotrophs in the forests. By analogy, this is akin to stating that the size of bird nests and number of eggs in the bird nests is a significant portion of ecosystem production -- this isn’t to say that bird nests and eggs, nor macrofungal reproductive structures are unimportant. But the evidence of how these measures are connected to ecosystem processes needs to be there, and in this document, as written, we only have a set of five reproduction-related morphological traits of these species.

Rather, their measures of “functional” traits are adequate for what they have defined (lines 105-109) as “response” traits (“quantifiable ...reproductive characteristics...related to fitness” -- the latter term I take to refer to the evolutionary type of fitness connected to reproductive output) -- mushroom cap size, size and shape of spores are all aspects of macrofungal reproduction. As far as I know, it would be a stretch to say that larger pilei and larger spores are reliable measures of ecosystem productivity.

In its current state the manuscript provides no direct measure or evidence of “effect” traits (“how species contribute to ecosystem functionality”) which, again, is typically considered as having roles in primary productivity, nutrient cycling, carbon storage, etc. As written, the manuscript does not adequately address how these data are related to these “effect” traits and only gives us the “response” of a few traits of macrofungal reproduction. Nor has the manuscript sufficiently explained how measuring only these five morphological traits we can get to a “functional” understanding of forests and the impact of urbanization on these forests by way of fungal responses. Perhaps there is some way to explain how these five traits connect to the functional view, but as is, the manuscript does not do so convincingly.

To address these concerns, I make two suggestions. First, throughout the manuscript including the title, consider using the term “morphology-related trait” instead of “functional” to more clearly show what was measured. The authors could still make the contention that this trait set is an element of “functional diversity” but repeated use of what was actually measured would consistently remind readers that only morphology and not some ecosystem process (e.g. decomposition or carbon sequestration) was considered by this study. Given this, they should minimize their extrapolation of these data to speaking to ecosystem processes.

Second, it is not out of the question to include some actual “effect” trait data in this manuscript that could be related to the urbanization gradient. Although I suggested this in the previous comments, the rebuttal to this comment was not clear [How is classifying a species of fungi as ectomycorrhizal or saprotrophic “subjective”?] and this new manuscript has not yet adequately addressed the point of concern. How would it not be a part of this study to address “effect” type functional traits, especially if it wouldn’t take much more effort (and no collection of additional data) to do so? If I am missing something here, then it needs to be clarified in the manuscript as readers will share my confusion.

One way to do this with the data already collected and shared (see Supplemental File S4) is to use the simple model provided in Avis et al (2016, Restoration Ecology, see below) and calculate an “effect” type functional value trait index for each of their sites. To do this, each taxon listed in S4 can be scored as ectomycorrhizal, saprotrophic or parasitic, and then by using the formula below, an “effect” type functional value can be calculated for each plot. These data can then be compared and statistically analyzed across the urbanization gradient. This really wouldn’t be much more work and from the sounds of it in the reply to this comment, probably related to your species richness patterns.

The Simple Model of Avis et al 2016 (Restoration Ecology)
(1) Simple Model Site Simple = [ FEcM × ln(#speciesEcM + 1 ] +[ FSap × ln(#speciesSap + 1 ] +[ FPar × ln(#speciesPar + 1 ] Where F is functional value as defined in Table S3b, EcM is EcM fungi, Sap is saprotrophic fungi, and Par is parasitic fungi.

With the suggested approach above, the authors can easily test their contention in their rebuttal [“We think that using trophic groups for studies on functional diversity is not appropriate due to very dissimilar species are grouped together based only on one trait (trophic strategy) and a lot of information is missing.’] In doing so, they can greatly expand the relevance of their work.


Additional line by line comments.

Line 71: Change “decreasing continuum” to “decrease of human influence…”
Line 119 and through the manuscript: Should “true” be “taxonomic”? If not, then other diversity, for example, “functional diversity” would be “false”.
Lines 176-177: I’m not sure what “practical purposes” would prevent changing “individual” to “collection unit” throughout the paper
Lines 183-185: This sentence can be removed as it goes without saying.
Line 187: State the index used in relationship to the variables measured, i.e. what Jost formula was used and what were the input variables, species observed? This is particularly important to do since no true knowledge or evidence exists as to how many actual genetic “individuals” were observed in this study. How a “Jost” approach might incorporate that consideration into “true diversity” should be considered as well.

Reviewer 2 ·

Basic reporting

No comment

Experimental design

No comment

Validity of the findings

No comment

Additional comments

No comments

---

## Round 0.3 · accepted · Accept

Thank you for fully addressing the additional reviewers' comments.